# Temporal-spatial variability of modern climate in the Altai Mountains during 1970-2015

Yinbo Li[1], Dongliang Zhang[2,3]*, Mariia Andreeva[4], Yaoming Li[2,3], Lianlian Fan[2,3], Min Tang[5]

**1** College of Resource and Environmental Science, Xinjiang University, Urumqi, China, **2** State Key Laboratory of Desert and Oasis Ecology, Chinese Academy of Sciences, Xinjiang Institute of Ecology and Geography, Urumqi, China, **3** Chinese Academy of Sciences, Research Center for Ecology and Environment of Central Asia, Urumqi, China, **4** St. Petersburg State University, St. Petersburg, Russia, **5** Altai Branch of the Altai National Forestry Administration of China, Aletai, China

* zhdl@ms.xjb.ac.cn

**Data Availability Statement:** The data from Russia were obtained from the website (http://meteo.ru/english/data/). The data from China were downloaded from the website (http://data.cma.cn/).

## Abstract

Located in the intermediate zone between the taiga forests in Siberian Plain and the deserts in Central Asia, the Altai Mountains are of scientific concern about Holocene climate change in the past decades. However, researches about modern climate changes are relatively scarce in the Altai Mountains. In this study, temporal- spatial changes of air temperature and precipitation were investigated systematically in the Altai Mountains based on fifteen meteorological records over the period of 1970–2015. The Altai Mountains experienced a rapid warming trend with a rate of 0.41˚C/decade and an insignificantly wetting trend at a rate of 4.82 mm/decade during 1970–2015. The magnitude of temperature trend was negatively correlated with elevation in cold season (spring and winter), whereas that was positively correlated with elevation in warm season (summer and autumn). The cyclonic anomalies to the northwest and an anticyclonic anomalies to the southeast blocked the southward cold air and then provided the favorable condition for an increasing precipitation via the southwesternly wind in the Altai Mountains.

## Introduction

The AR5 of IPCC proposes a significantly warming trend during the past century on earth [1]. There is growing evidence that the rate of warming is amplified with elevation, such that high-elevation environments experience more rapid changes in temperature than environments at low elevation [2–5]. The Asian Central Arid Zone (ACAZ) is rightly situated in the core of Eurasia. Due to few water vapor supplies from the near oceans (i.e., the North Atlantic Ocean, the Pacific Ocean and the Indian Ocean), glaciers and snow in high-elevation regions of ACAZ (i.e., the Tianshan Mountains and the Altai Mountains) become the most important solid water reservoir and support the oasis agriculture developments and urban operations. However, they are in a state of rapid melting heavily affected by the climate warming, leading to the water resources at risk [6–10]. It means that the temporal-spatial temperature and precipitation variability in ACAZ should be paid more attention on. The changes of modern temperature and precipitation in the Tianshan Mountains, the largest mountain system in ACAZ,

**Funding:** This research was financially supported by Strategic Priority Research Program of Chinese Academy of Sciences (No. XDA20020101), by China Postdoctoral Science Foundation (No. 2019M663864), and by the Western Young Scholar Program-B of Chinese Academy of Sciences (No. 2018-XBQNXZ-B-020). The primary analysis of Russian meteorological data was done with support of Russian Science Foundation (No. 17-77-10041). The funders had no role in study design, data collection and analysis, decision to publish, or preparation of the manuscript.

**Competing interests:** The authors declare no competing interests. The funders had no role in the design of the study; in the collection, analyses, or interpretation of data; in the writing of the manuscript; or in the decision to publish the results.

have been detailedly investigated [11–18]. The increasing trends were revealed in mean annual temperature (MAT) [17–18] and mean annual precipitation (MAP) [14, 16] in the Tianshan Mountains over the past decades.

The Altai Mountains (Fig 1A), the second largest mountain system within ACAZ and located in the north of the Tianshan Mountains, stretch across Russia, Mongolia, China and Kazakhstan. They are of scientific concern in term of paleoclimatic studies for understanding the interaction between the westerlies and the Asian summer monsoon and for exploring the associations between climate modulation and cultural evolution along the 'Eurasian Steppe Belt' [19–25]. Compared with the detailed investigations about modern climate changes in the Tianshan Mountains, modern climate changes are poorly analyzed in the Altai Mountains. For example, Malygina et al. [26] detected the precipitation changes in the Russian Altai and in the Mongolian Altai in the interval spanning from 1959 to 2014 and estimated the possible driving factors. Zhang et al. [27] based on a converse moisture character during the Holocene interval and analyzed the different time-scale (i.e., season, year, multi-decades and centennial/millennial scales) climatic changes in the Russian Altai and in the Chinese Altai. It is clear that modern climate trends were not systematically investigated in terms of different sub-regions and of different elevations in the Altai Mountains.

Based on the selected meteorological data in the Altai Mountains, we detailedly investigated the temporal-spatial changes of modern climate of the Altai Mountains during 1970–2015. This study is very important for assessing the influences of climate changes on its physical environments and ecosystem stabilities or securities and for improving the abilities for ecological management in the whole Altai Mountains.

## Regional setting

Located in the transition zone between the Siberian taiga forests and the Central Asian deserts, the Altai Mountains stand out from its surrounding territories with their unique diversity of

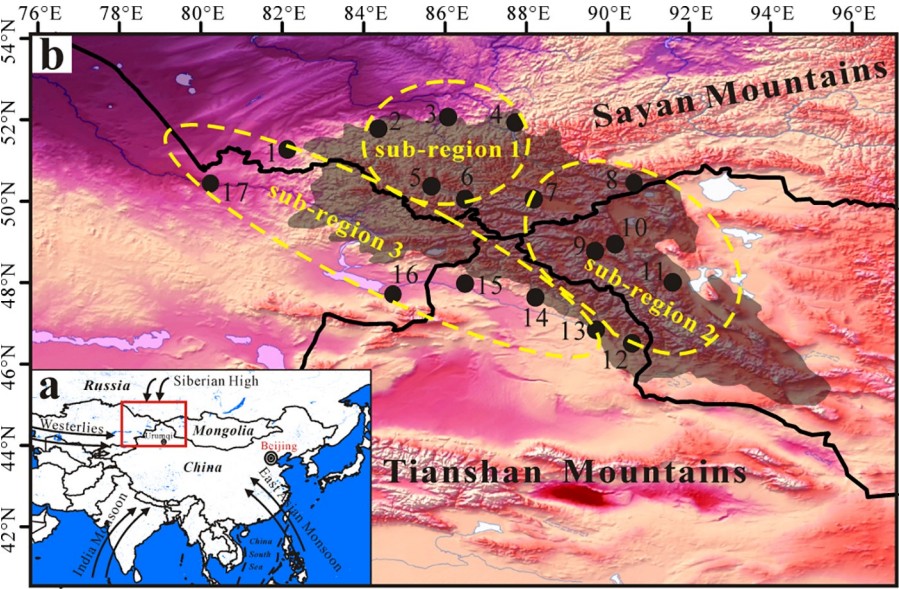

**Fig 1.** Geographic location of the Altai Mountains (a) and the related meteorological stations in the Altai Mountains (white area) (b). The Altai Mountains are divided into three sub-regions (1, 2 and 3) and the detailed information of meteorological stations is showed in Table 1. The image was downloaded from the Natural Earth website (http://www.naturalearthdata.com/ ).

**Table 1. Related information of meteorological stations in the Altai Mountains.**

| No. | Station Name | Latitude (˚N) | Longitude (˚E) | Altitude (m) | Annual temperature (˚C) | Annual precipitation (mm) | Interval |
|---|---|---|---|---|---|---|---|
| 1 | Zmeinogorsk | 51.15 | 82.17 | 354 | 2.81 | 692.78 | 1966–2016 |
| 2 | Soloneshnoe | 51.63 | 84.33 | 409 | 1.86 | 581.20 | 1966–2016 |
| 3 | Kyzyl-Ozek | 51.90 | 86.00 | 331 | 2.07 | 745.00 | 1966–2016 |
| 4 | Yailu | 51.77 | 87.60 | 480 | 3.73 | 894.95 | 1966–2016 |
| 5 | Mugur-Aksy | 50.38 | 90.43 | 1850 | -2.36 | 142.60 | 1966–2016 |
| 6 | Ust-Coksa | 50.30 | 85.60 | 978 | 0.62 | 472.23 | 1940–2017 |
| 7 | Kara-Tyurek | 50.00 | 86.40 | 2600 | -5.54 | 601.41 | 1940–2017 |
| 8 | Kosh-Agach | 50.00 | 88.04 | 1760 | -4.96 | 119.93 | 1936–2017 |
| 9 | Yalalt | 48.80 | 89.50 | 2148 | 2.75 | 134.16 | 1970–2015 |
| 10 | Olgiy | 48.96 | 89.98 | 1715 | 1.23 | 111.08 | 1961–2015 |
| 11 | Khovd | 48.08 | 91.35 | 1405 | 1.24 | 132.85 | 1961–2015 |
| 12 | Qinghe | 46.67 | 90.38 | 1220 | 0.72 | 125.01 | 1958–2017 |
| 13 | Fuyun | 46.98 | 89.52 | 826.6 | 2.88 | 177.71 | 1962–2017 |
| 14 | Aletai | 47.73 | 88.08 | 736.9 | 4.48 | 190.13 | 1958–2017 |
| 15 | Habahe | 48.05 | 86.40 | 534 | 4.77 | 199.93 | 1958–2017 |
| 16 | Zajsan | 47.47 | 84.92 | 603 | 4.12 | 294.53 | 1967–2000 |
| 17 | Semipalatinsk | 50.42 | 80.30 | 196 | 3.58 | 285.66 | 1940–2011 |

ecosystems [28–31]. They have the largest unbroken stretches of Siberian fir, pine and larch trees in the world. The particularly remarkable species is dark coniferous taiga. Glaciers mainly occupy in the central part of Altai Mountains, especially on the ranges of Katunsky, Taldurinsky, Ak-Turu, Munku-Sardyk, South and North Chuisky, Kryzin [32–37]. They give life to two largest rivers (e.g., Ob and Yenisei river) and several large lakes (e.g., Khuvsgul, Uvs, Teletskoe and Wulungu lake). The climate in the Altai Mountains is featured by the harsh continental climate [31–37]. Due to the most of humid air masses transported by the westerlies from the west throughout a year, the Altai Mountains are featured by a strong northwest to southeast precipitation gradient [38–39]. The Siberian High controls the cold-season climate and prevents precipitation formation in the Altai Mountains [26–27, 38, 40].

## Data resource and method

### Data resource

The selected stations were showed in the Altai Mountains in Fig 1 and their detailed information was showed in Table 1. The eight stations are situated in the Russian Altai Mountains and they are Zmeinogorsk, Soloneshnoe, Kyzyl-Ozek, Yailu, Mugur-Aksy, Ust-Coksa, Kara-Tyurek and Kosh-Agach. The three stations are located in the Mongolian Altai Mountains and they are Olgyi, Yalalt, Khovd. The four stations are situated in the Chinese Altai Mountains and they are Habahe, Aletai, Fuyun and Qinghe. The data from Russia was obtained from the website (http://meteo.ru/english/data/) and the data from Mongolia was provided by Dr. Mariia Andreeva. The data from China was downloaded from the website (http://cma.gov.cn). These meteorological data from three countries was both pre-disposed through the strict quality control and homogenized. The data from Semipalatinsk and Zajsan meteorological station within Kazakhstan was selected and was just for calculating seasonal features of temperature and precipitation because of no long time-scale observed data. The seasonal changes were also taken into account and spring is from March to May, summer is from June to August, autumn is from September to November, and winter is from December to February in the next year.

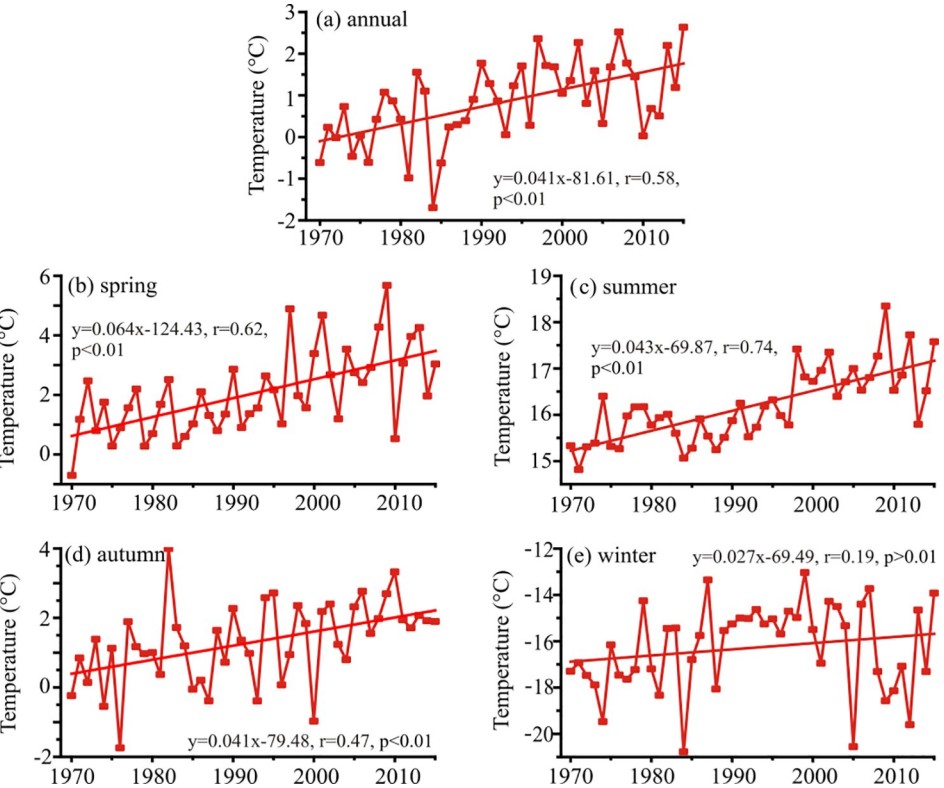

**Fig 2. Variations of annual and seasonal mean temperature in the Altai Mountains during 1970–2015.**

## Method

The treated method (i.e., the Mann-Kendall trend test) was applied to probe the temperature and precipitation trends during the observed period [16, 38–42]. If the slope value is >0, the climate data has a positive trend and the climate data shows a negative trend when the slope value is <0. The data of slope value indicates the rate or the magnitude at which the climate data increases or decreases.

## Results

### Temperature trend characters

The temperature variations in annual and seasonal scale of the Altai Mountains were showed in Fig 2. The result shows that MAT experienced an increasing trend during the studied interval (1970–2015), which had a significance level of 0.01. The increased rate was 0.41˚C/decade in the Altai Mountains during 1970–2015. In the studied interval, about 1.85˚C was the value which the MAT increased (Fig 2A). A significant increase of air temperature was also showed in spring, summer and autumn (Fig 2B–2D), whereas air temperature in winter was statistically insignificant (Fig 2E). Their increased rates were 0.64˚C/decade for spring, 0.43˚C/decade for summer, 0.41˚C/decade for autumn and 0.27˚C/decade for winter, respectively (Fig 2B–2E). The rising rate of seasonal temperature was the quickest in spring.

Fig 3 presented the spatial divergent rates of MAT and seasonal temperature in the Altai Mountains. The significantly positive trends (p<0.05) were observed in MAT among all fifteen stations. The MAT were divergent and the value spanning from 0.28˚C/decade in

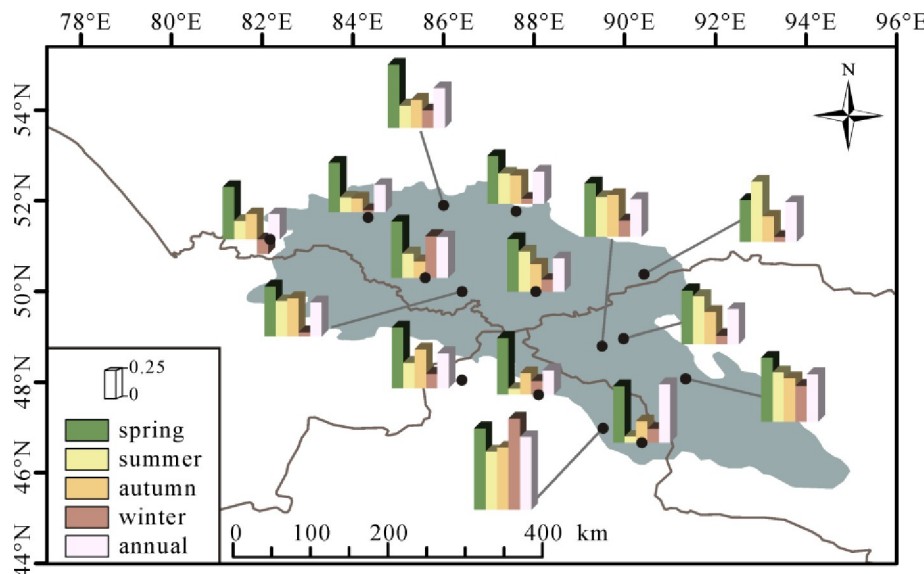

**Fig 3. Spatial trends for annual and seasonal temperature during 1970–2015 over the Altai Mountains.**

Zmeinogorsk to 0.75˚C/decade in Fuyun between 1970 and 2015. The rates in majority of stations fall spanning from 0.20 to 0.40˚C/decade, that in four stations waved between 0.40 and 0.60˚C/decade, and that in two stations fluctuated between 0.60 and 0.80˚C/decade. The fastest increased rate of MAT was observed at Fuyun and Qinghe in the southeastern Altai Mountains. All stations significantly increased (from 0.43 to 0.84˚C/decade) in spring, larger than the increased trend of MAT. The rising rates in summer varied spanning from 0.05 (Altai and Qinghe) to 0.61 (Fuyun)˚C/decade, while the insignificant changes of some stations (i.e., Zmeinogorsk, Soloneshnoe, Altai and Qinghe) were observed. In autumn, except Soloneshnoe (0.04˚C/decade, P>0.05) station, others showed a warming rate ranging from 0.20 to 0.65˚C/decade. In winter, a decreasing trend of temperature (−0.12˚C/decade) with no significance was detected at Zmeinogorsk. The winter air temperature at Ust-Coksa, Fuyun and Khovd experienced increasing trends which exceeded the 95% confidence level. The rates insignificantly increased between 0.02 and 0.20˚C/decade in other stations. As a whole, the temporal and spatial rates of temperature were varied in annual and seasonal scale and the warming climate was found in the whole Altai Mountains during the past decades.

## Precipitation trend characteristics

Fig 4 showed the trends of MAP and seasonal precipitation during 1970–2015 in the Altai Mountains. An increased rate about 4.82 mm/decade (P>0.05) was detected for MAP (Fig 4A). The MAP in 2015 increased 21.69 mm comparing with that in 1970. The changes of seasonal precipitation all did not exceed the significance level of 0.05 and their rates were 2.10 mm/decade in spring, 1.60 mm/decade in summer, 0.25 mm/decade in autumn and 0.93 mm/decade in winter, respectively (Fig 4B–4E).

The spatial changes of precipitation in annual and seasonal scale were presented in Fig 5. Four stations (Zmeinogorsk, Kyzyl-Ozek, Kosh-Agach and Olgiy) had decreasing trends of MAP with significant levels (P<0.05) and their rates were −4.68, −8.66, −7.29 and −4.99 mm/decade, respectively. Although the positive trends of MAP were detected among other stations, the significant trends were only found in Soloneshnoe, Habahe, Aletai, Fuyun and Qinghe and their rates ranged from 1.14 (Khovd) to 17.97 (Soloneshnoe) mm/decade. In spring, two

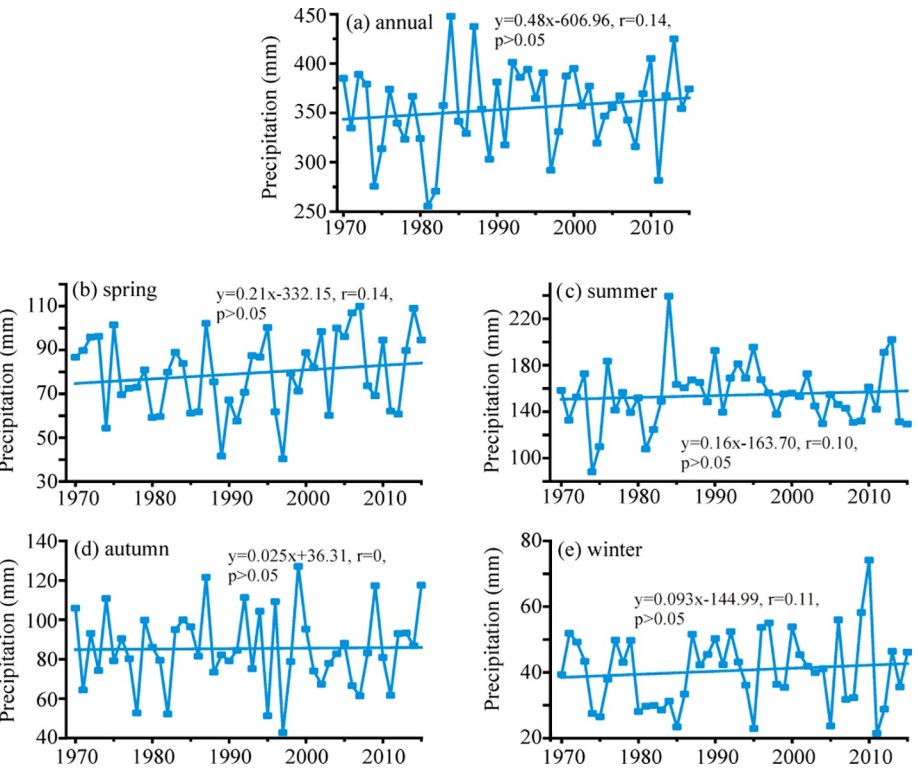

**Fig 4. Variations of annual and seasonal mean precipitation in the Altai Mountains during 1970–2015.**

stations (Zmeinogorsk and Kosh-Agach) exhibited decreasing trends with no significance. Other remaining stations showed increasing trends with rates spanning from 0.16 to 6.40 mm/ decade, but most of them did not past 95% confidence level. In summer, five stations (Mugur-Aksy, Kara-Tyurek, Yailu, Olgiy and Khovd) revealed declining trends with no significance.

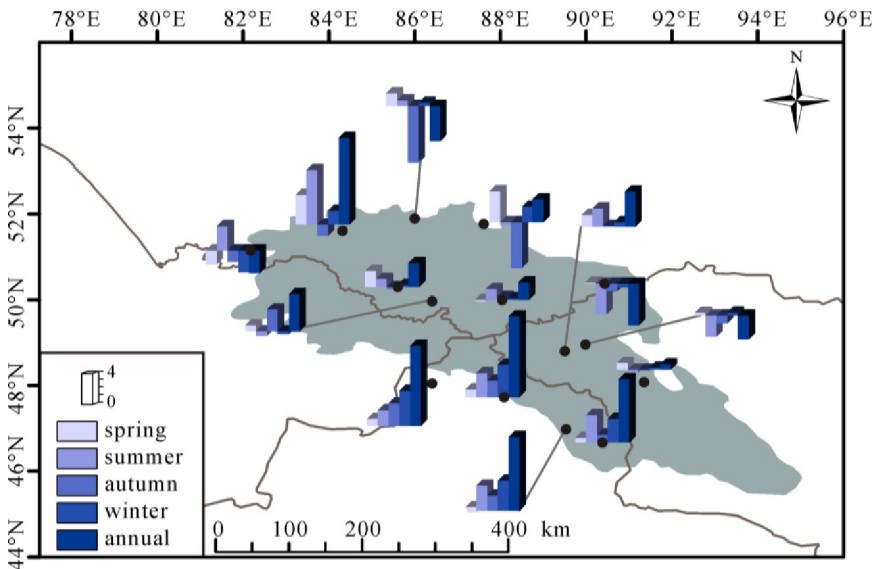

**Fig 5. Spatial trends for annual and seasonal precipitation during 1970–2015 over the Altai Mountains.**

Except Soloneshnoe, the precipitation at other remaining nine stations was featured by insignificant climbing trends (P>0.05). The increasing amplitude in Soloneshnoe was the most evident and the largest with a rate of 11.27 mm/decade. The autumn precipitation exhibited insignificantly decreasing trends among eight stations in the northern Altai and in the eastern Altai. Insignificant trends were also observed in other stations with increasing rates spanning from 0.14 (Yalalt) to 4.71 (Habahe) mm/decade. In winter, only Zmeinogorsk, Mugur-Aksy and Kara-Tyurek showed decreasing trends with insignificant levels (P>0.05). All stations in the southern Altai Mountains had a significant increasing trend of precipitation in winter, while no significance was found among other stations. Overall, the Altai Mountains experienced a slightly wetting trend since 1970, especially in the southern Altai.

## Evolution of temperature and precipitation in sub-regions

The monthly temperature in the Altai Mountains is characterized by the highest temperature in June-August and the lowest temperature in November-January during 1970–2015 (Fig 6A). The highest temperature (22.97˚C) was recorded in Zajsan and the lowest value (-28.78˚C) was in Kosh-Agach (Fig 6A). The three features of monthly precipitation were characterized in the Altai Mountains (Fig 6B). Firstly, the monthly precipitation was unimodal and was mainly concentrated in warm season (summer and autumn) with 55–84%. The related stations are Soloneshnoe, Kyzyl-Ozek, Yailu, Kara-Tyurek and Ust-Coksa. The MAP in these stations was relatively high (average about 649.73 mm). Secondly, the feature of monthly precipitation was also unimodal, but the MAP was relatively low (average about 133.86 mm). The associated stations are Mugur-Aksy, Kosh-Agach within Russia, three stations within Mongolia and Qinghe within China. Thirdly, being different to the former two features of precipitation, the distribution of monthly precipitation was bimodal characterized by two peaks at April-September (50–68%) and at November- December (13–21%). The associated stations include Zmeinogorsk, Habahe, Aletai, Fuyun, Semipalatinsk and Zajsan.

Based on the features of monthly precipitation, the Altai Mountains was classified into three sub-regions: sub-region 1, 2 and 3 (Fig 1). The related trends of MAT and MAP among three sub-regions during 1970–2015 were presented in Fig 7. The result suggests that MAT all significantly increased among three sub-regions (Fig 7A). The most rapidly increased

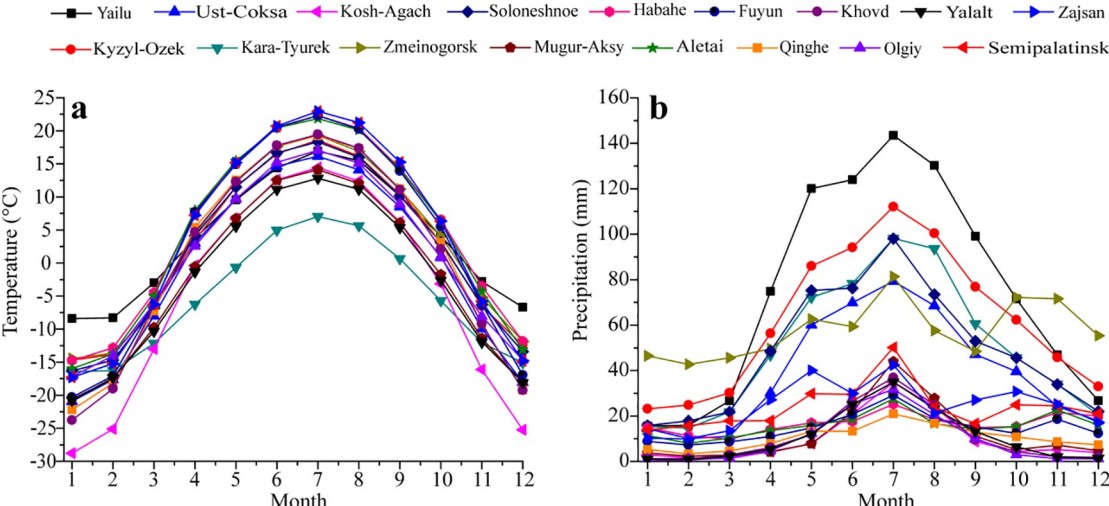

**Fig 6.** Variations of monthly temperature (a) and monthly precipitation (b) in the Altai Mountains during 1970–2015.

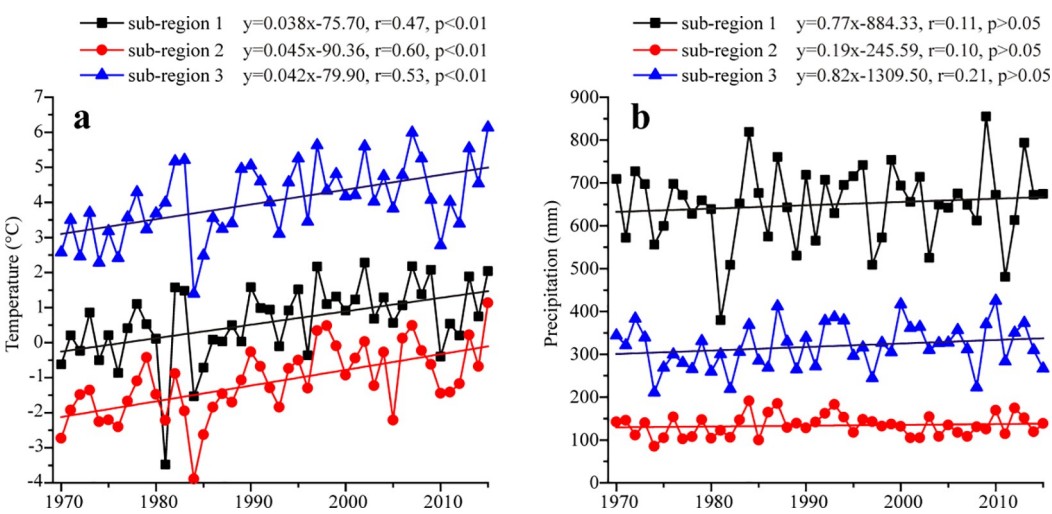

**Fig 7.** Variations of mean annual temperature (a) and mean annual precipitation (b) in three sub-regions of the Altai Mountains during 1970–2015.

temperature occurred in sub-region 2 at a rate of 0.45˚C/decade. The next was in sub-region 3 at a rate of 0.42˚C/decade and the third in sub-region 1 at 0.38˚C/decade. Being different to the temperature fluctuations, the trends of MAP were insignificantly and their rates were 7.7 mm/decade in sub-region 1, 1.9 mm/decade in sub-region 2 and 8.2 mm/decade in sub-region 3, respectively (Fig 7B).

## Temperature/precipitation trend-elevation relationship

We further analyzed the temperature/precipitation trends in different elevations (i.e., high elevation >2000 m, middle elevation at 2000–1000 m and low elevation <1000 m) (Fig 8). The stations in high elevation are Kara-Tyurek and Yalalt. Mugur-Aksy, Kosh-Agach, Olgiy, Khovd and Qinghe are contained in middle elevation and the remaining stations are included in low elevation. A significant warming during 1970–2015 with different rates was found in

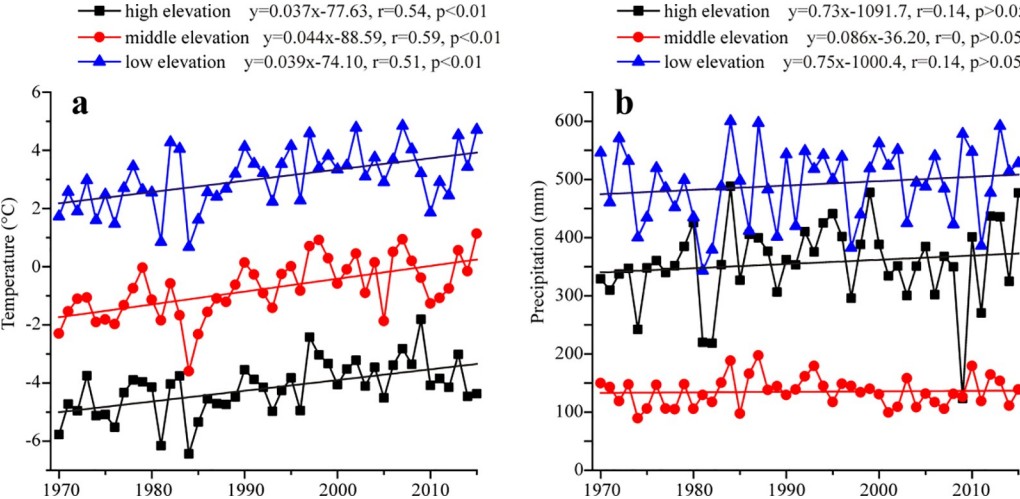

**Fig 8.** Variations of mean annual temperature (a) and mean annual precipitation (b) in different elevations of the Altai Mountains during 1970–2015.

different elevations of the Altai Mountains (Fig 8A). Their increased rates were 0.37°C/decade in high elevation, 0.44°C/decade in middle elevation and 0.39°C/decade in low elevation, respectively.

Being consistent with the synchronously increasing trend of MAT in different elevations, the MAP in different elevations was also increasing in the studied period (Fig 8B) and their rates were 7.27 mm/decade in high elevation, 0.86 mm/decade in middle elevation and 7.49 mm/decade in low elevation, respectively. All trends of MAP were insignificant (P>0.05). Overall, we found that the variations of MAP were complex than that of MAT in different sub-regions and different elevations of the Altai Mountains, being similar with their changes in the Tianshan Mountains [9]. It should be noted that the the fastest warming rightly corresponds to the smallest increased precipitation in sub-region 2 and in middle elevations.

## Discussions

### Analysis of divergent trends of climate change and elevations

It is very important to reveal the relationship between temperature and elevation for understanding regional climate response to global warming. Two proposals about relationships between temperature trend and elevation were existed. Firstly, the warming rate in higher elevations was much larger than that in lower elevations [4–7, 43–45]. Secondly, the divergent magnitude of temperature trend was not associated with elevation [9, 46]. The relationships between the magnitude of MAT and seasonal temperature trend and elevation in the Altai Mountains were showed in Fig 9. The slight positive correlations between MAT trend magnitude and elevation were found in annual, summer and autumn, the relationship was only significant in summer. The negative relationship occurred in spring and winter, and they were not significant. Overall, the magnitude of temperature trend was negatively correlated with

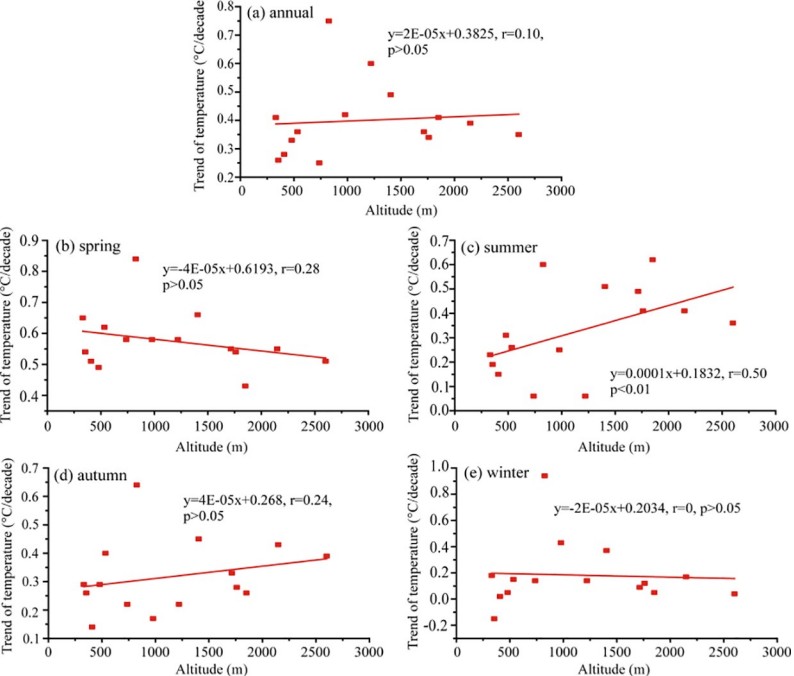

**Fig 9. Relationships between annual and seasonal air temperature trend magnitude and elevation in the Altai Mountains.**

elevation in cold season (spring and winter), whereas that was positively correlated with elevation in warm season (summer and autumn), being similar with that in the Tianshan Mountains [9]. This elevation-dependent temperature depends upon the changes of snow cover and surface albedo feedback in the Altai Mountains [9, 47–48]. In details, the larger snow cover and its stronger albedo feedback in cold season accelerate the transit of upward turbulent heat, the cooling air has a lower probability of being heated by latent heat in high elevation than that in low elevation. In warm season, the decreased snow cover and the weakening surface albedo can not inhibit the warming air in high elevation, i.e., high-elevation environments experience more rapid changes in temperature than environments at low elevations [2–5].

Furthermore, understanding the elevation dependent wetting or not is also of significance. Previous studies revealed precipitation increased with elevation climbing in the mountains on earth [49–51]. The similar wetting trend with elevation was also observed in the adjacent Tianshan Mountains [9, 16]. The relationships between MAP and seasonal precipitation trend magnitude and elevation in the Altai Mountains were showed in Fig 10. There were slight negative correlations between precipitation trend magnitude and elevation for annual, spring, summer and winter, and they were insignificant. The only positive relationship occurred in autumn. The strong elevation dependency of seasonal precipitation appeared in autumn with a correlation efficient of 0.23 which was statistically insignificant at the 95% confidence level. The autumn precipitation from 1970 to 2015 displayed an increasing tendency at a rate of 0.9 mm/decade each 1000 m. It suggests larger changes of autumn precipitation in low elevation than in high elevation of the Altai Mountains during 1970–2015. Being inconsistent with the wetting trend with elevation in summer and autumn in the Tianshan Mountains [9, 16], the elevation dependency of MAP and seasonal precipitation trends are more intricate in the Altai Mountains.

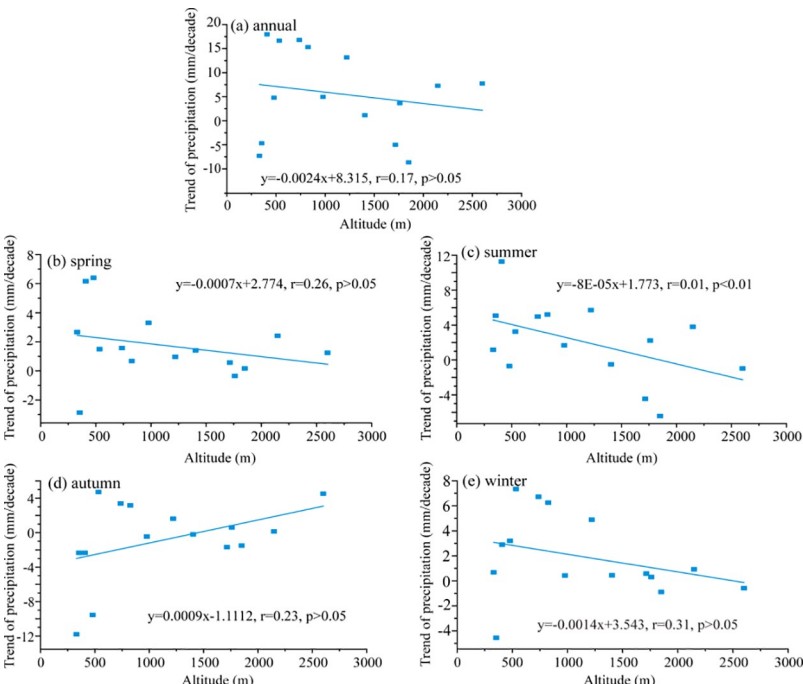

**Fig 10. Relationships between annual and seasonal precipitation trend magnitude and elevation in the Altai Mountains.**

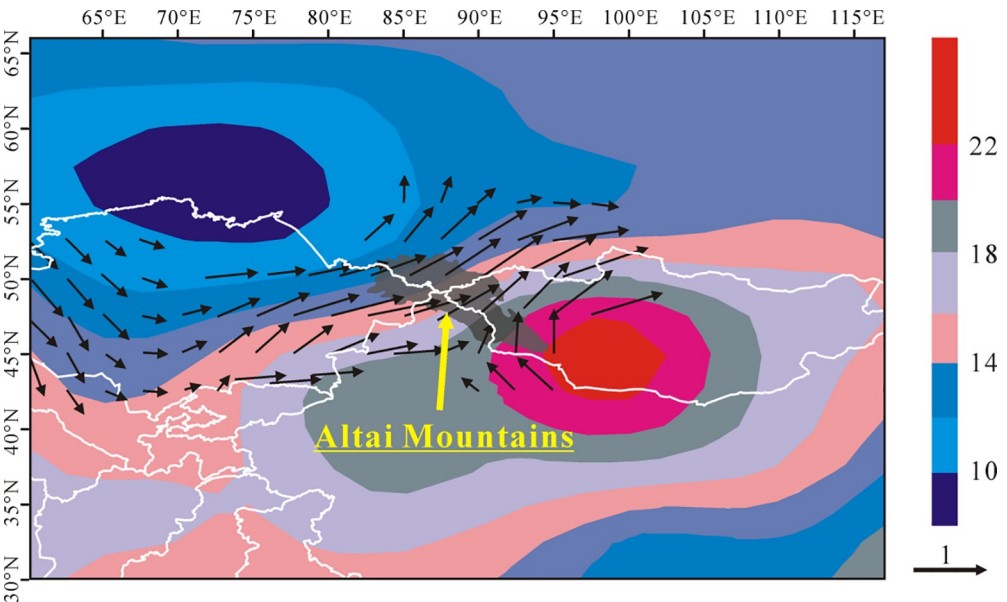

**Fig 11. Differences of geopotential height (shaded) and wind speed (vector) at 500 hPa between 1996–2016 and 1960–1995 (modified from Fig 12 in Xu et al. (2018)).** The length and direction of arrows mean the flux and the direction of wind.

## Association between atmospheric circulations and climate information

The temperature trends in the Altai Mountains were similar with that in the Northern Hemisphere in recent decades and both experienced a warming climate. The rising rate of temperature was the quickest in spring, which was attributable to the substantially declined snow cover in the past 30 years, especially in early spring through summer interval [52–54]. Fossil fuel and biofuel emissions of black carbon plus organic matter were mainly responsible for spring-time snow cover loss over Eurasia including the Altai Mountains [55]. Although three features of monthly precipitation were existed in the Altai Mountains, MAP was featured by an insignificantly increasing trend, which is inconsistent with the wetting Tianshan Mountains [9].

Regional precipitation variability in the Altai Mountains could be closely related with atmosphere circulation. Xu et al. [9] calculated that the differences of geopotential height (shaded) and wind speed (vector) at 500 hPa in the Asian Central Arid Zone including the Altai Mountains through calculating the circulation using NCEP/NCAR reanalysis data (Fig 11). As shown in Fig 11, the Altai Mountains were controlled by a cyclonic anomalies to their northwest and an anticyclonic anomalies to their southeast. This kind of atmospheric modes blocked the southward cold air and in turn provided favorable environment for water vapor transport via the westerlies from the North Atlantic, the Mediterranean and Caspian seas to the Altai Mountains [56–57]. The horizontal moisture advection and wind convergence terms has a significant positive contribution to the wetting trend in the northwest China [58]. The southwesternly wind between the anomaly cyclone and the anomaly anticyclone driven the water vapor flux from the Mediterranean and Caspian sea into the Altai Mountains, which resulted in an increasing precipitation. Compared with the precipitation changes in sub-region 2, the larger increased precipitation in sub-region 1 and 3 could result from the effect of windward slope [27].

The combination of complex topography and other local effects may be responsible for different relationships between precipitation trend and elevation in the Altai Mountains [16–17].

Several uncertainties can influence the relationship of precipitation trend with elevation in mountainous regions over recent decades. Firstly, the long-term observed data of precipitation are extremely sparse at high elevations. For example, only two stations (Kara-Tyurek and Yalalt) are above 2000 m and no observed data is above 3000 m in the Altai Mountains. Meanwhile, there is no available long-term precipitation data in the Altai Mountains within Kazakhstan. The CRU (Climatic Research Unit) and GPCC (Global Precipitation Climatology Centre) data are able to describe the temporal-spatial variations of precipitation, but their uncertainties limit their applicability in the mountainous areas [59–60]. Secondly, there is a lack of consistency in the observation instruments and environments, such as vegetation changes, human grazing and relocation of stations because the Altai Mountains cover over four countries (China, Kazakhstan, Russia and Mongolia). The uncertainties would influence the relationships between climate and elevation to a certain extent, and the climate-elevation relationships should be observed and be investigated in future through (1) establishing more high-elevation stations and (2) combining three main methods including surface in-situ climate observations, satellite remote-sensing data and high-resolution climatic modeling [2–5, 16].

## Conclusions and implications

The Altai Mountains experienced a rapid warming trend with a rate of 0.41˚C/decade and an insignificantly wetting trend at a rate of 4.82 mm/decade during 1970–2015. Being inconsistent with the precipitation increasing trend with elevation in the Tianshan Mountains over recent decades [9, 16], the insignificantly increasing trend was showed in different elevations. The magnitude of temperature trend was negatively correlated with elevation in cold season (spring and winter), whereas that was positively correlated with elevation in warm season (summer and autumn). Additionally, the trends of precipitation magnitude with elevation in the Altai Mountains were complex. Two warning signs we should focus on. The first is that no obvious increasing precipitation poses a potential threat to the regional forest developments in middle elevations under the warming condition. The second is that no obvious increasing precipitation directly affects storage of water resource in the Altai Mountains under the warming condition. In addition with significant mass loss of glaciers in the Altai Mountains [36–37, 61], a decreasing water storage may be a heavy threat to human survival through impacts on water availability. This urge us to take action to protect the safety of ecology and water resource in the Altai Mountains.

## Supporting information

**S1 Data.**
(RAR)

## Author Contributions

**Data curation:** Yinbo Li, Mariia Andreeva, Lianlian Fan.

**Funding acquisition:** Dongliang Zhang.

**Investigation:** Dongliang Zhang, Yaoming Li, Min Tang.

**Project administration:** Mariia Andreeva.

**Resources:** Mariia Andreeva, Yaoming Li.

**Writing – original draft:** Yinbo Li, Dongliang Zhang, Lianlian Fan, Min Tang.

**Writing – review & editing:** Dongliang Zhang.

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
