## [Decision Letter · Decision Letter 0]

14 Nov 2019

PONE-D-19-22736

Temporal-spatial variability of modern climate in the Altai Mountains during 1970-2015

PLOS ONE

Dear Dr. Zhang,

Thank you for submitting your manuscript to PLOS ONE. After careful consideration, we feel that it has merit but does not fully meet PLOS ONE’s publication criteria as it currently stands. Therefore, we invite you to submit a revised version of the manuscript that addresses the points raised during the review process.

One reviewer raised serious concerns, and thought this manuscript is similar with the paper published in Scientific Reports from the same research team in 2018. Therefore, I would suggest you have to address this concern carefully. Specific attention should also be paid as follows: 

You suggested that this anticyclone mode blocked the water vapour transport from the west. Why does this weakened water vapour transport not cause a drying trend? Why does the strengthening of Siberian High in winter not cause a cooling in the Altai Mountains?

We would appreciate receiving your revised manuscript by Dec 29 2019 11:59PM. To enhance the reproducibility of your results, we recommend that if applicable you deposit your laboratory protocols in protocols.io, where a protocol can be assigned its own identifier (DOI) such that it can be cited independently in the future. For instructions see: http://journals.plos.org/plosone/s/submission-guidelines#loc-laboratory-protocols

We look forward to receiving your revised manuscript.

Kind regards,

Bao Yang, Ph.D, Prof.

Academic Editor

PLOS ONE

Journal Requirements:

1. Thank you for including your competing interests statement; "No"

2. We note that [Figure(s) 1] in your submission contain [map/satellite] images which may be copyrighted. All PLOS content is published under the Creative Commons Attribution License (CC BY 4.0), which means that the manuscript, images, and Supporting Information files will be freely available online, and any third party is permitted to access, download, copy, distribute, and use these materials in any way, even commercially, with proper attribution. For these reasons, we cannot publish previously copyrighted maps or satellite images created using proprietary data, such as Google software (Google Maps, Street View, and Earth). For more information, see our copyright guidelines: http://journals.plos.org/plosone/s/licenses-and-copyright.

1.    You may seek permission from the original copyright holder of Figure(s) [1] to publish the content specifically under the CC BY 4.0 license. 

4. Please include your tables as part of your main manuscript and remove the individual files. Please note that supplementary tables (should remain/ be uploaded) as separate "supporting information" files

5. Thank you for including your funding statement; "No"

Please provide an amended Funding Statement that declares *all* the funding or sources of support received during this specific study (whether external or internal to your organization) as detailed online in our guide for authors at http://journals.plos.org/plosone/s/submit-now.  

Please state what role the funders took in the study.  If any authors received a salary from any of your funders, please state which authors and which funder. If the funders had no role, please state: "The funders had no role in study design, data collection and analysis, decision to publish, or preparation of the manuscript."

Reviewers' comments:

Reviewer's Responses to Questions

**Comments to the Author**

1. Is the manuscript technically sound, and do the data support the conclusions?

Reviewer #1: Yes

Reviewer #2: Partly

Reviewer #3: Partly

Reviewer #4: Partly

2. Has the statistical analysis been performed appropriately and rigorously? 

Reviewer #1: Yes

Reviewer #2: Yes

Reviewer #3: No

Reviewer #4: No

3. Have the authors made all data underlying the findings in their manuscript fully available?

Reviewer #1: Yes

Reviewer #2: Yes

Reviewer #3: Yes

Reviewer #4: No

4. Is the manuscript presented in an intelligible fashion and written in standard English?

Reviewer #1: Yes

Reviewer #2: Yes

Reviewer #3: No

Reviewer #4: Yes

5. Review Comments to the Author

Reviewer #1: The reviewer congratulates the authors for this extensive analysis of “Temporal-spatial variability of modern climate in the Altai Mountains during 1970-2015”. This is a noble piece of work and will be very useful for the scientific community. This will provide the first hand information regarding the climate of Altai mountains. The comments are attached in the attachment.

Reviewer #2: Review of paper titled “Temporal-spatial variability of modern climate in the Altai Mountains during 1970-2015” by Li et al.

General comment: accepted after minor revisions

In this paper, the author investigated the air temperature and precipitation trends in Altai Mountains based on 15 meteorological records over the period 1970-2015. Significant temperature increasing trends and insignificant precipitation trends were found over the Altai Mountains, with increased rate of precipitation trend and decreased rate of temperature trend. These trends were attributed to the enhancing anticyclone circulation and increasing geopotential height.

This study is straightforward and meaningful, climate changes over the high-elevation regions and arid regions, and their societal and economic influences are concerned by both research community and general public. However, I do have some concerns about the analyses and results.

For the introduction section, I believe some recent literatures about climate changes over the high-elevation regions, would also be helpful to strengthen the motivation and importance of this manuscript, such as:

Pepin, N., R. S. Bradley, H. F. Diaz, et al., 2015: Elevation-dependent warming in mountain regions of the world. Nature - Climate Change, 5, 424-430, DOI: 10.1038/NCLIMATE2563

Diaz, H. F., R. S. Bradley, and L. Ning, 2014: Climatic changes in mountain regions of the American Cordillera and the tropics: historical changes and future outlook. Arctic Antarctic & Alpine Research, 46(4), 735-743.

Bradley, R. S., F. T. Keimig, H. F. Diaz, and D. R. Hardy, 2009: Recent changes in freezing level heights in the tropics with implications for the deglacierization of high mountain regions. Geophys. Res. Lett., 36, L17701, doi: 10.1029/2009GL037712

Bradley, R. S., M. Vuille, H. F. Diaz, and W. Vergara, 2006: Threats to water supplies in the tropical Andes. Science, 312, 1755-1756

Line 137, any physical explanations about the largest increasing trend in spring?

The three sub-regions used in Fig. 7 should be marked in a map.

In Fig. 8, which stations are categorized as high-, middle-, low-elevations?

About the section of “temperature/precipitation trend-elevation relationship”, first, I am not surprised about no significant relationship between precipitation trends and elevations. But, for the relationship between temperature trends and elevation, usually larger increasing trends over high-elevation regions are more acceptable to the whole community, because of several potential mechanisms: snow albedo, water vapor changes, and aerosols etc. (Pepin et al., 2015). In Fig. 8, 0.037 C/decade, 0.044 C/decade, and 0.039C/decade are not very different, especially how many stations in each category are not given. Therefore, I would recommend the authors use another data (e.g., CRU data) to to repeat this analysis and confirm their results.

In the following discussion, the authors tried to use the snow cover to explain the converse relationships between temperature trends and elevations. But, I think seasonal snow cover extents can only be used to explain the climatology, rather than the changes. While, the changes of snow cover and surface albedo feedback will support the elevation dependent warming. Deeper discussion about changes of snow cover and corresponding mechanisms is needed in the revision.

In the section of “association between atmospheric circulations and climate information”, when using correlations between precipitation changes and large-scale climate variability to explain the precipitation changes, the authors mixed the concepts of inter-annual variations and trends. Meanwhile, AO, NAO, and ENSO are inter-annual scale climate variability, while AMO and PDO are decadal scale climate variability, so they influence precipitation variability on different scales. Therefore, the authors need to clarify which of these two changes (or both) they want to discuss, then discuss the corresponding mechanisms. Moreover, the mechanisms behind influences from the AO, NAO, and AMO are easy to understand through the water vapor transportation through the westerly. While, the authors should provide more details about mechanisms behind influences from PDO and ENSO to the precipitation over the Altai Mountains.

In Fig. 11, why the whole period is divided into 1996-2016 and 1960-1995? Any sudden change or shift around 1995/96?

Maybe I missed some files, but I did not see Table 1 and Table 2.

Caption of Fig. 10, should it be mean annual precipitation?

Reviewer #3: General comments:

This study analyzed the temporal-spatial variability of precipitation and temperature in the Altai Mountains during 1970-2015. Meanwhile, the authors intend to provide interpretations for the climate variations. However, the interpretations are improper. The English is not really good, and the paper would greatly benefit from a thorough language editing.

Detailed comments:

1.Lines 35-36: Why does the warming result in the wetting in mid-latitude regions?

2.Lines 37-38: Revise “the mid-latitude regions” to “Eurasia”.

3.Where are the tables?

4.Line 108: AMO and PDO are not atmosphere circulation indices.

5.Lines 107-116: Why do you select these five climate indices, which cannot explain the results in this study as analyzed in the Discussions of this paper?

6.Lines 113-116: Why do you unconventionally divide the seasons?

7.Lines 212-218: It is better to display the sub-regions in Figure 1.

8.Lines 279-281: Figure 10c indicates larger changes in lower elevations than that in the higher elevations.

9.Lines 285-299: The correlations reflect the interannual relationships between the indices and precipitation in the Altai Mountains. However, this study focused on the decadal variability of precipitation in the Altai Mountains. Hence, this analysis is improper for this study.

10.Lines 300-315: This study explored the climate changes during 1970-2015, whereas Figure 11 analyzed the circulation changes from 1960 to 2016. Hence, this analysis is also improper for this study.

11.Lines 285-315: The first paragraph suggests that global circulations have weak relationship with the precipitation. The second paragraph indicates that local circulation might modulate the influence of global circulations on precipitation. However, the detailed mechanism is not proposed in this study.

The authors suggest that this anticyclone mode blocked the water vapour transport from the west. Why does this weakened water vapour transport not cause a drying trend? Why does the strengthening of Siberian High in winter not cause a cooling in the Altai Mountains?

Figure 11 indicates that the Altai Mountains are controlled by cyclonic anomalies to their northwest and an cyclonic anomalies to their southeast, rather than an anticyclone circulation.

11. Fig. 10: Revise “air temperature” to “precipitation”.

Reviewer #4: The manuscript attempted to discuss spatiotemporal characteristics of the modern climate, specifically precipitation and temperature, in the Altai Mountains and associated atmospheric circulations. This research is important and meaningful. The results reveal that the Altai Mountains experience a rapid warming, while no significant trend was found. The Altai Mountains was divided into 3 subregions based on the characters of monthly precipitation, and the trends of annual and seasonal precipitation and temperature were also studied for each subregion. However, the method used to divide these subregions is not very convincing. Meanwhile, the explanation about the atmospheric circulations associated with the climate variability over the Altai Mountains is not clear, and even contradictory to the observed trends. Furthermore, this manuscript is similar with the paper published in Scientific Reports from the same research team in 2018. Therefore, I suggest that this manuscript can’t be accepted by this journal.

General comments:

1. Tables that mentioned in the manuscript are missing.

2. The trends of annual and seasonal mean temperature/precipitation for the fifteen stations in the Altai Mountains have been investigated and compared with each other. A table containing this information can be added to make readers easier to follow. In the analysis of trend-elevation relationship, which stations are included in the high elevations, middle elevations, and low elevations need to be clarified.

3. The three subregions are divided based on the three characters of monthly precipitation. For the northern Altai Mountains (subregion 1), the monthly precipitation in Zmeinogorsk station shows two peaks (Fig.6), but the other five stations in this sub-region all shows one peak. Why is the Zmeinogorsk station included in this subregion? More explanation is needed. At the same time, I have the similar concern about the southern Altai Mountains (subregion 3). This subregion includes Qinghe, Aletai, Fuyun, and Habaha stations. The monthly precipitation of Qinghe station shows one peak, but two peaks can be found in other three stations. ¬¬The division of sub-regions need more justifications.

4. For the atmospheric circulations responsible for the climate variability in the Altai Mountains is not clear. This manuscript stated that the Altai Mountains experienced a rapid warming (line 318, line 22), but it also argued that no rapid warming in the Altai Mountains due to the strengthening of Siberian High in winter (lines 308-309). This is contradictory. Meanwhile, how does the anticyclone mode block less water vapor transported by the westerlies from the North Atlantic Ocean into the Altai? I don’t think the reference 35 talks about it.

Specific comments:

1. Lines 125: the subtitle “Temperature and precipitation during 1970-2015” is redundant.

2. There is a small figure on the left bottom of figure 1. The description about it needs to be added to the caption of figure 1

3. For figure 3, there is a small histogram and ”0.47” on the top of the legend. What does that stand for? Figure 5 has the same problem.

4. The figure 6 doesn’t have “a” on the temperature (left figure) and “b” on the precipitation (right figure)

5. For the caption of figure 8, change “annual mean precipitation” to” mean annual precipitation”

6. The “seasonal air temperature” needs to be added to the caption of figure 9 and figure 10.

7. In figure 10, the left string (annual, summer, etc.) covered the data point.

5. The Altai Mountains has been divided into 3 subregions (northern, eastern and southern). Showing these subregions in a figure will be clearer. Maybe the division of subregions can be added to the figure 1.

6. PLOS authors have the option to publish the peer review history of their article (what does this mean?). If published, this will include your full peer review and any attached files.

Reviewer #1: No

Reviewer #2: No

Reviewer #3: No

Reviewer #4: No

---

## [Author Response · Author response to Decision Letter 0]

15 Dec 2019

Plos One (PONE-D-19-22736)

Dear Professor Yang Bao: 

Thanks very much for taking your and the reviewers’ time to review this manuscript. We really appreciate all your comments and suggestions! Please find our responses in below and my revisions/corrections in the re-submitted files. 

Thank you very much for your attention and consideration again.

Sincerely yours

Zhang Dongliang

2019-12-11

Replies to Editor (thank you!):

One reviewer raised serious concerns, and thought this manuscript is similar with the paper published in Scientific Reports from the same research team in 2018. Therefore, I would suggest you have to address this concern carefully. Specific attention should also be paid as follows: 

Reply: Yes, we published the paper in Scientific Reports in 2018. That paper revealed the out-of- phase relationship of precipitation changes at different time-scales (i.e., season, year, multi-decades, centennial and millennial scales). In this study, we systematically investigated the temporal-spatial changes of modern climate in the Altai Mountains during 1970-2015, including temperature and precipitation trends and their changes in different sub-regions, temperature and precipitation trend-elevation relationship and possible reasons. 

Replies to Reviewer 1 (thank you!):

Line 78: I think it is species not ‘specie’

Reply: Modified.

Provide a better figure 2. The years at the bottom is not clear.

Reply: Added.

Line 169-170: please rephrase “Around 21.69 mm of annual mean precipitation was increased from 1970 to 2015”.

Reply: Done. 

This MAP in 2015 increased 21.69 mm comparing with that in 1970. 

Line 176: “with significant levels:. You need to mention the value.

Reply: Added.

Line 190: change ‘trends was’ to ‘trends were’

Reply: Added.

Line 198: please change the heading to “mean annual cycle or temporal evolution of temperature and precipitation”.

Reply: Thank you. We changed the heading to “evolution of temperature and precipitation in sub-regions”.

The tables are missing from the manuscript. Please provide. 

Reply: Added.

The elevation dependent temperature depends upon the various factor, such as snow-albedo feedback, nocturnal cloud, change in surface energy balance, aerosol and change in water vapour and fluxes. The authors must look into these to confirm which mechanism dominates over Altai mountains in deciding the elevation dependent warming.

Reply: Done.

This character should result from the lakes (e.g., Wulungu Lake, Zaysan Lake, Teleskoyel Lake and Achit Lake) in low-elevation regions of the Altai Mountains, which significantly alter the surface energy and then influence on regional climate via mediating the fluxes of energy, moisture and momentum [9, 47-48]. In details, during the warm season (summer and autumn), the lakes absorb incoming solar radiation and inhibit upward turbulent heat transport, the humid air has a lower probability of being heated by latent heat released from condensation at low elevations which induce cooling and lower trend in low elevation. During the cold season (spring and winter), the air is warmed and showed a greater trend magnitudes in low elevations due to the heat is released from lakes.

Replies to Reviewer 2 (thank you!):

1, For the introduction section, I believe some recent literatures about climate changes over the high-elevation regions, would also be helpful to strengthen the motivation and importance of this manuscript, such as:

Reply: We added the several papers you suggested.

2, Line 137, any physical explanations about the largest increasing trend in spring?

Reply: Thank you for your question. We investigated the possible causes.

The rising rate of temperature was the quickest in spring, which was attributable to the substantially declined snow cover inferred from satellites in the past 30 years, especially in early spring through summer interval (Zhai and Zhou, 1997; Qian et al., 2011; Huang et al., 2012). Fossil fuel and biofuel emissions of black carbon plus organic matter were mainly responsible for springtime snow cover loss over Eurasia (Flanner et al., 2009). 

Huang JP, Guan XD, Ji F. Enhanced cold-season warming in semi-arid regions. Atmospheric Chemistry and Physics, 2012; 12(12): 5391-5398.

3, The three sub-regions used in Fig. 7 should be marked in a map.

Reply: Thank you! We showed these sub-regions in Figure 1b.

4, In Fig. 8, which stations are categorized as high-, middle-, low-elevations?

Reply: Thank you! We added them in the revised manuscript.

We further analyzed the temperature/precipitation trends in different elevations. The Altai Mountains were divided into three gradients (i.e., >2000 m, 1000-2000 m and <1000 m) (Fig. 8). The stations within >2000 m were Kara-Tyurek and Yalalt. Mugur-Aksy, Kosh-Agach, Olgiy, Khovd and Qinghe were contained elevation between 2000 and 1000 m and the remaining stations were considered elevation <1000 m. 

5, About the section of “temperature/precipitation trend-elevation relationship”, first, I am not surprised about no significant relationship between precipitation trends and elevations. But, for the relationship between temperature trends and elevation, usually larger increasing trends over high-elevation regions are more acceptable to the whole community, because of several potential mechanisms: snow albedo, water vapor changes, and aerosols etc. (Pepin et al., 2015). In Fig. 8, 0.037 C/decade, 0.044 C/decade, and 0.039C/decade are not very different, especially how many stations in each category are not given. Therefore, I would recommend the authors use another data (e.g., CRU data) to to repeat this analysis and confirm their results.

Reply: Thank you. Your suggestion is very valuable. According to your suggestion, we checked CRU and GPCC data. The study showed that both CRU and GPCC products are able to describe the temporal-spatial variation of precipitation in China. However, the CRU data showed large biases in the Tibetan Plateau and some large-mountain areas including the Tianshan Mountains and the Altai Mountains (Wang and Wang, 2017). The study also showed that GPCC is a better choice compared to CRU for studying the long-term precipitation trend in China (Wang and Wang, 2017). Although GPCC agrees well with observed data, the uncertainty of the satellite rainfall products limits their applicability in mountainous areas (Jin et al., 2016). Therefore, we pay more attention on the observed data in the Altai Mountains.

Wang D, Wang AH. Applicability assessment of GPCC and CRU precipitation products in China during 1901 to 2013. Climatic and Environmental Research, 2017; 22(4): 446-462 (in Chinese).

Jin XL, Shao H, Zhang C, Yan Y. The Applicability Evaluation of Three Satellite Products in Tianshan Mountains. Journal of Natural Resources, 2016; 31(12): 2074-2085 (in Chinese).

6, In the following discussion, the authors tried to use the snow cover to explain the converse relationships between temperature trends and elevations. But, I think seasonal snow cover extents can only be used to explain the climatology, rather than the changes. While, the changes of snow cover and surface albedo feedback will support the elevation dependent warming. Deeper discussion about changes of snow cover and corresponding mechanisms is needed in the revision.

Reply: Done.

This character should result from the lakes (e.g., Wulungu Lake, Zaysan Lake, Teleskoyel Lake and Achit Lake) in low-elevation regions of the Altai Mountains, which significantly alter the surface energy and then influence on regional climate via mediating the fluxes of energy, moisture and momentum [9, 47-48]. In details, during the warm season (summer and autumn), the lakes absorb incoming solar radiation and inhibit upward turbulent heat transport, the humid air has a lower probability of being heated by latent heat released from condensation at low elevations which induce cooling and lower trend in low elevation. During the cold season (spring and winter), the air is warmed and showed a greater trend magnitudes in low elevations due to the heat is released from lakes.

7, In the section of “association between atmospheric circulations and climate information”, when using correlations between precipitation changes and large-scale climate variability to explain the precipitation changes, the authors mixed the concepts of inter-annual variations and trends. Meanwhile, AO, NAO, and ENSO are inter-annual scale climate variability, while AMO and PDO are decadal scale climate variability, so they influence precipitation variability on different scales. Therefore, the authors need to clarify which of these two changes (or both) they want to discuss, then discuss the corresponding mechanisms. Moreover, the mechanisms behind influences from the AO, NAO, and AMO are easy to understand through the water vapor transportation through the westerly. While, the authors should provide more details about mechanisms behind influences from PDO and ENSO to the precipitation over the Altai Mountains.

Reply: Thank you very much. According to your valuable suggestion, we didn’t discuss the associations between precipitation and driving factors (including AO, NAO, ENSO and so on) because no obvious correlations were found in the revised manuscript. In this study, we pay more attention on the effect of atmospheric circulations on the climate changes of the Altai Mountains.

As shown in Fig. 11, the Altai Mountains were controlled by cyclonic anomalies to their northwest and an anticyclonic anomalies to their southeast. This kind of atmospheric modes blocked the northward cold air and in turn provided favorable environment for an increasing temperature in the Altai Mountains. The water vapor from the North Atlantic, the Mediterranean and Caspian sea was transported to the Altai Mountains via the westerlies (Aizen et al., 2001, 2006). Previous study indicated that the sum of horizontal moisture advection and wind convergence terms has a significant positive contribution to the wetting trend in the northwest region of China (Peng and Zhou, 2017). As shown in Fig. 11, the southwestern wind between the anomaly cyclone and the anomaly anticyclone driven the water vapor flux from the Mediterranean and Caspian sea into the Altai Mountains, which in turn, increases the precipitation. The larger increased precipitation appeared in sub-region 1 and 3 during 1970-2015 due to the effect of windward slope with a significant change in southern Altai Mountains within China [27].

8, In Fig. 11, why the whole period is divided into 1996-2016 and 1960-1995? Any sudden change or shift around 1995/96?

Reply: Thank you for your question. Yes, you are right. The mutations of MAT occurred in 1995 and the large atmospheric circulation influenced on the mutations of climate. This result was inferred from the analysis of temperature in the nearby Tianshan Mountains by Xu et al. (2018).

9, Maybe I missed some files, but I did not see Table 1 and Table 2.

Reply: We are sorry we missed them in the original manuscript. Now we added the table in the resubmitted manuscript.

10, Caption of Fig. 10, should it be mean annual precipitation?

Reply: Done.

Replies to Reviewer 3 (thank you!):

1. Lines 35-36: Why does the warming result in the wetting in mid-latitude regions?

Reply: Thank you. Sorry, we made a mistaken and modified it. The AR5 of IPCC (2013) pointed out that in the 21st century global warming will further intensify the Earth’s water cycles, making the high-latitude areas even wetter and mid- and low-latitude areas even drier, melting more glaciers and reducing spring snow covers in the Northern Hemisphere. 

Intergovernmental Panel on Climate Change Fifth Assessment Report (IPCC AR5). Summary for Policymakers: The Physical Science Basis, Contribution of Working Group I to the IPCC Fifth Assessment Report Climate Change, 2013. 

2. Lines 37-38: Revise “the mid-latitude regions” to “Eurasia”.

Reply: Done.

3. Where are the tables?

Reply: Added.

4. Line 108: AMO and PDO are not atmosphere circulation indices.

Reply: Deleted.

5. Lines 107-116: Why do you select these five climate indices, which cannot explain the results in this study as analyzed in the Discussions of this paper?

Reply: Thank you for your question. Based on no obvious correlation between five indices and climate changes, we deleted this part. We focused on the analysis of association between atmospheric circulations and climate information.

6. Lines 113-116: Why do you unconventionally divide the seasons?

Reply: Modified and reanalyzed data. 

Winter, spring, summer and autumn are defined as extending from December to the following February (DJF), March to May (MAM), June to August (JJA) and September to November (SON), respectively.

7. Lines 212-218: It is better to display the sub-regions in Figure 1.

Reply: Displayed.

8. Lines 279-281: Figure 10c indicates larger changes in lower elevations than that in the higher elevations.

Reply: Modified.

9.Lines 285-299: The correlations reflect the interannual relationships between the indices and precipitation in the Altai Mountains. However, this study focused on the decadal variability of precipitation in the Altai Mountains. Hence, this analysis is improper for this study.

Reply: Thank you for your valuable suggestion. We deleted the correlation analysis and paid more attention on association between atmospheric circulations and climate information.

10.Lines 300-315: This study explored the climate changes during 1970-2015, whereas Figure 11 analyzed the circulation changes from 1960 to 2016. Hence, this analysis is also improper for this study.

Reply: Thank you. We referenced the study between atmospheric circulations and climate changes during 1960-2016 in the nearby Tianshan Mountains, which was finished by Xu et al. (2018).

11. Lines 285-315: The first paragraph suggests that global circulations have weak relationship with the precipitation. The second paragraph indicates that local circulation might modulate the influence of global circulations on precipitation. However, the detailed mechanism is not proposed in this study.

Reply: Thank you. We paid more attention on association between atmospheric circulations and climate information.

The authors suggest that this anticyclone mode blocked the water vapour transport from the west. Why does this weakened water vapour transport not cause a drying trend? Why does the strengthening of Siberian High in winter not cause a cooling in the Altai Mountains?

Reply: Thank you. We modified it. 

As shown in Fig. 11, the Altai Mountains were controlled by cyclonic anomalies to their northwest and an anticyclonic anomalies to their southeast. This kind of atmospheric modes blocked the northward cold air and in turn provided favorable environment for an increasing temperature in the Altai Mountains.

Figure 11 indicates that the Altai Mountains are controlled by cyclonic anomalies to their northwest and an cyclonic anomalies to their southeast, rather than an anticyclone circulation.

Reply: According to your suggestion, we modified it. 

As shown in Fig. 11, the Altai Mountains were controlled by cyclonic anomalies to their northwest and an anticyclonic anomalies to their southeast. This kind of atmospheric modes blocked the northward cold air and in turn provided favorable environment for an increasing temperature in the Altai Mountains.

11. Fig. 10: Revise “air temperature” to “precipitation”.

Reply: Changed.

Replies to Reviewer 4 (thank you ! ):

1. Tables that mentioned in the manuscript are missing.

Reply: Added.

2. The trends of annual and seasonal mean temperature/precipitation for the fifteen stations in the Altai Mountains have been investigated and compared with each other. A table containing this information can be added to make readers easier to follow. In the analysis of trend-elevation relationship, which stations are included in the high elevations, middle elevations, and low elevations need to be clarified.

Reply: Thank you! We added them.

We further analyzed the temperature/precipitation trends in different elevations. The Altai Mountains were divided into three gradients (i.e., >2000 m, 1000-2000 m and <1000 m) (Fig. 8). The stations within >2000 m were Kara-Tyurek and Yalalt. Mugur-Aksy, Kosh-Agach, Olgiy, Khovd and Qinghe were contained elevation between 2000 and 1000 m and the remaining stations were considered elevation <1000 m. 

3. The three subregions are divided based on the three characters of monthly precipitation. For the northern Altai Mountains (subregion 1), the monthly precipitation in Zmeinogorsk station shows two peaks (Fig. 6), but the other five stations in this sub-region all shows one peak. Why is the Zmeinogorsk station included in this subregion? More explanation is needed. At the same time, I have the similar concern about the southern Altai Mountains (subregion 3). This subregion includes Qinghe, Aletai, Fuyun, and Habaha stations. The monthly precipitation of Qinghe station shows one peak, but two peaks can be found in other three stations. The division of sub-regions need more justifications.

Reply: Thank you very much for pointing out the mistake. We did not notice it. Based on your suggestion, we redivided the sub-regions.

Fig. 6 showed monthly temperature and monthly precipitation changes of the Altai Mountains during 1970-2015. The highest temperature appears in June-August and the lowest temperature in November-January. The highest temperature (22.97 °C) in summer was recorded in Zajsan and the lowest value (-28.78 °C) in winter was in Kosh-Agach (Fig. 6a). Three features of monthly precipitation are characterized in the Altai Mountains (Fig. 6b). Firstly, the monthly precipitation is unimodal and is mainly concentrated in warm season (summer and autumn) with 55-84%. The related stations are Soloneshnoe, Kyzyl-Ozek, Yailu, Kara-Tyurek and Ust-Coksa. Their MAP is also relatively abundant (average about 649.73 mm). Secondly, the feature of monthly precipitation is also unimodal, but the MAP is relatively low (average about 133.86 mm). The associated stations are Mugur-Aksy, Kosh-Agach within Russia, three stations in Mongolia and Qinghe in China. Thirdly, different to the former two features of precipitation, the distribution of monthly precipitation is bimodal characterized by two peaks at April-September (50-68%) and at November-December (13-21%). The associated stations include Zmeinogorsk in Russia, Habahe, Aletai and Fuyun in China, Semipalatinsk and Zajsan in Kazakhstan.

4. For the atmospheric circulations responsible for the climate variability in the Altai Mountains is not clear. This manuscript stated that the Altai Mountains experienced a rapid warming (line 318, line 22), but it also argued that no rapid warming in the Altai Mountains due to the strengthening of Siberian High in winter (lines 308-309). This is contradictory. Meanwhile, how does the anticyclone mode block less water vapor transported by the westerlies from the North Atlantic Ocean into the Altai? I don’t think the reference 35 talks about it.

Reply: Thank you very much. We rewritten them.

Regional precipitation variability in climate trends could be closely related with atmosphere circulation. Xu et al. [9] calculated that differences of geopotential height (shaded) and wind speed (vector) at 500 hPa in the Asian Central Arid Zone including the Altai Mountains through calculating the circulation using NCEP/NCAR reanalysis data (Fig. 11). As shown in Fig. 11, the Altai Mountains are controlled by cyclonic anomalies to their northwest and an anticyclonic anomalies to their southeast. This kind of atmospheric modes blocked the northward cold air and in turn provided favorable environment for an increasing temperature in the Altai Mountains. The water vapor from the North Atlantic, the Mediterranean and Caspian sea is transported to the Altai Mountains via the westerlies (Aizen et al., 2001, 2006). Previous study indicated that the sum of horizontal moisture advection and wind convergence terms has a significant positive contribution to the wetting trend in the northwest region of China (Peng and Zhou, 2017). As shown in Fig. 11, the southwestern wind between the anomaly cyclone and the anomaly anticyclone drives the water vapor flux from the Mediterranean and Caspian sea into the Altai Mountains, which resulted in an increasing precipitation. The larger increased precipitation appeared in sub-region 1 and 3 due to the effect of windward slope with a significant change in the southern Altai Mountains within China [27].

Specific comments:

1. Lines 125: the subtitle “Temperature and precipitation during 1970-2015” is redundant.

Reply: Deleted.

2. There is a small figure on the left bottom of figure 1. The description about it needs to be added to the caption of figure 1.

Reply: Added.

3. For figure 3, there is a small histogram and ”0.47” on the top of the legend. What does that stand for? Figure 5 has the same problem.

Reply: Thank you very much. We modified them. The numbers in a small histogram of Figure 3 and 5 indicate the changeable values of temperature and precipitation.

4. The figure 6 doesn’t have “a” on the temperature (left figure) and “b” on the precipitation (right figure)

Reply: Added.

5. For the caption of figure 8, change “annual mean precipitation” to” mean annual precipitation”

Reply: Modified.

6. The “seasonal air temperature” needs to be added to the caption of figure 9 and figure 10.

Reply: Added.

7. In figure 10, the left string (annual, summer, etc.) covered the data point.

Reply: Modified.

5. The Altai Mountains has been divided into 3 subregions (northern, eastern and southern). Showing these subregions in a figure will be clearer. Maybe the division of subregions can be added to the figure 1.

Reply: Added.

---

## [Decision Letter · Decision Letter 1]

15 Jan 2020

PONE-D-19-22736R1

Temporal-spatial variability of modern climate in the Altai Mountains during 1970-2015

PLOS ONE

Dear Dr. Zhang,

Thank you for submitting your manuscript to PLOS ONE. After careful consideration, we feel that it has merit but does not fully meet PLOS ONE’s publication criteria as it currently stands. Therefore, we invite you to submit a revised version of the manuscript that addresses the points raised during the review process.

ACADEMIC EDITOR: 

One reviewer suggested that it still needs major revising, and I agree. Please revised it carefully. Additionally I would recommend that the text be polished carefully.

We would appreciate receiving your revised manuscript by Feb 29 2020 11:59PM. To enhance the reproducibility of your results, we recommend that if applicable you deposit your laboratory protocols in protocols.io, where a protocol can be assigned its own identifier (DOI) such that it can be cited independently in the future. For instructions see: http://journals.plos.org/plosone/s/submission-guidelines#loc-laboratory-protocols

We look forward to receiving your revised manuscript.

Kind regards,

Bao Yang, Ph.D, Prof.

Academic Editor

PLOS ONE

Additional Editor Comments (if provided):

This manuscript has been revised greatly. However, the reviewer suggested that it still needs revising, and I agree. There are some wording errors in the text additionally. I would recommend that the text be polished carefully.

Reviewers' comments:

Reviewer's Responses to Questions

**Comments to the Author**

1. If the authors have adequately addressed your comments raised in a previous round of review and you feel that this manuscript is now acceptable for publication, you may indicate that here to bypass the “Comments to the Author” section, enter your conflict of interest statement in the “Confidential to Editor” section, and submit your "Accept" recommendation.

Reviewer #2: (No Response)

Reviewer #3: (No Response)

2. Is the manuscript technically sound, and do the data support the conclusions?

Reviewer #2: Yes

Reviewer #3: Yes

3. Has the statistical analysis been performed appropriately and rigorously? 

Reviewer #2: Yes

Reviewer #3: Yes

4. Have the authors made all data underlying the findings in their manuscript fully available?

Reviewer #2: (No Response)

Reviewer #3: Yes

5. Is the manuscript presented in an intelligible fashion and written in standard English?

Reviewer #2: (No Response)

Reviewer #3: Yes

6. Review Comments to the Author

Reviewer #2: The authors have successfully addressed all my concerns, and I can now recommend this manuscript to be accepted.

Reviewer #3: General comments:

This manuscript has remarkably improved compared with the original version. However, some further modifications should be required.

Detailed comments:

1.Lines 27-28: Revise “in the northwest” to “ to the northwest”, “in the southeast” to “ to the southeast”, and “northward” to “southward” (northward air is warm air, also in Line 296).

2.Lines 30 and 302: “southwestern” to “southwesterly”

3.Lines 45-46: “leading to put the security of water resources at risk” has syntax error.

4.Line 48 and other places in this paper: “divergent” is more appropriate than “changeable”.

5.Line 68: “detailed” to “detailedly”

6.Line 70: “are” to “is”

7.Fig. 2e: p>0.01

8.Lines 119-123 and Lines 153-157: The comparisons are meaningless because they are not in the same period. The trends would be different even in the same area between different periods.

9.Lines 127-128: “Fig. 2b-d” to “Fig. 2b-e”

10.Line 161: “Fig. 4b-d” to “Fig. 4b-e”

11.Line 209: “both” is inappropriate

12.Lines 247-259: The interpretation is not convincing. The lake areas are very small compared with the study area. The effect of the lake on the temperature is local and very weak. In addition, the lakes are freezing up in cold seasons, which would cool rather than warm the air via reflecting solar radiation.

13.Fig. 10b: p>0.01

14.Lines 271-273: Negative trend in lower elevation also indicates precipitation change, the range of which is larger than that in higher elevation.

15.Fig. 11: which season? In addition, the differences can be calculated by the authors according the study period rather than modifying from Xu et al. (2018).

16.Line 315: CRU and GPCC are not satellite-based data.

17.Lines 100-101 and Lines 318-320: In the Data Section, it is suggested that meteorological data is pre-disposed through strict quality control and homogenized. However, this data is doubted in the Discussion Section.

7. PLOS authors have the option to publish the peer review history of their article (what does this mean?). If published, this will include your full peer review and any attached files.

Reviewer #2: No

Reviewer #3: No

---

## [Author Response · Author response to Decision Letter 1]

10 Feb 2020

From: zhdl@ms.xjb.ac.cn

Authors replies to the review comments

Plos One (PONE-D-19-22736R1)

Dear Prof. Yang Bao: 

Thanks very much for taking your and the reviewers’ time to review this manuscript. We really appreciate all your comments and suggestions! Please find our responses in below and my revisions/corrections in the re-submitted files. 

Thank you very much for your attention and consideration again.

Sincerely yours

Zhang Dongliang

2020-2-10

Detailed comments:

1.Lines 27-28: Revise “in the northwest” to “ to the northwest”, “in the southeast” to “ to the southeast”, and “northward” to “southward” (northward air is warm air, also in Line 296).

Reply: Done.

2.Lines 30 and 302: “southwestern” to “southwesterly”

Reply: Done.

3.Lines 45-46: “leading to put the security of water resources at risk” has syntax error.

Reply: Done.

4.Line 48 and other places in this paper: “divergent” is more appropriate than “changeable”.

Reply: Done.

5.Line 68: “detailed” to “detailedly”

Reply: Done.

6.Line 70: “are” to “is”

Reply: Done.

7.Fig. 2e: p>0.01

Reply: Modified.

8.Lines 119-123 and Lines 153-157: The comparisons are meaningless because they are not in the same period. The trends would be different even in the same area between different periods.

Reply: Deleted.

9.Lines 127-128: “Fig. 2b-d” to “Fig. 2b-e”

Reply: Done.

10.Line 161: “Fig. 4b-d” to “Fig. 4b-e”

Reply: Done.

11.Line 209: “both” is inappropriate

Reply: Deleted.

12.Lines 247-259: The interpretation is not convincing. The lake areas are very small compared with the study area. The effect of the lake on the temperature is local and very weak. In addition, the lakes are freezing up in cold seasons, which would cool rather than warm the air via reflecting solar radiation.

Reply: Thank you very much. We neglect this important information about the freezing lakes in winter.

This elevation-dependent temperature depends upon the changes of snow cover and surface albedo feedback in the Altai Mountains [9, 47-48]. In details, the larger snow cover and its stronger albedo feedback in cold season accelerate the transit of upward turbulent heat, the cooling air has a lower probability of being heated by latent heat in high elevation than that in low elevation. In warm season, the decreased snow cover and the weakening surfce albedo can not inhibit the warming air in high elevation, i.e., high-elevation environments experience more rapid changes in temperature than environments at lower elevations [2-5].

13.Fig. 10b: p>0.01

Reply: Done.

14.Lines 271-273: Negative trend in lower elevation also indicates precipitation change, the range of which is larger than that in higher elevation.

Reply: Modified.

15.Fig. 11: which season? In addition, the differences can be calculated by the authors according the study period rather than modifying from Xu et al. (2018).

Reply: Thank you for your question. Fig. 11 shows the annual-scale differences of geopotential height (shaded) and wind speed (vector) at 500 hPa between 1996-2016 and 1960-1995. We referenced the Xu et al. (2018)’s results based on two reasons: firstly, the climate in the Tianshan Moutains and the Altai Mountains within the Central Asia are consistently influenced by the westerlies throughout a year and by the Siberian High in winter; secondly, the destination of this manuscript is from our study about Holocene climate change in the Altai Mountains. We recognized a delayed temperature increase at high elevations during the early Holocene, resulting in more humid conditions at high elevations compared to low elevations (Zhang and Feng, 2018). So we want to investigate the temporal and spatial change of climate in the observed interval. Honestly, we have difficulty in doing that. Thank you very much.

16.Line 315: CRU and GPCC are not satellite-based data.

Reply: We deleted the word “satellite”.

17.Lines 100-101 and Lines 318-320: In the Data Section, it is suggested that meteorological data is pre-disposed through strict quality control and homogenized. However, this data is doubted in the Discussion Section. 

Reply: Thank you for your question. Indeed, the collected data from stations in the Altai Mountains is pre-disposed through strict quality control and homogenized. In the Discussion section, we doubted the fact that there has been a lack of consistency in the observation instrument and environments, such as vegetation changes, human grazing and relocation of stations because the Altai Mountains stretch over four countries (Kazakhstan, China, Mongolia and Russia).

---

## [Decision Letter · Decision Letter 2]

25 Feb 2020

Temporal-spatial variability of modern climate in the Altai Mountains during 1970-2015

PONE-D-19-22736R2

Dear Dr. Zhang Dongliang,

We are pleased to inform you that your manuscript has been judged scientifically suitable for publication and will be formally accepted for publication once it complies with all outstanding technical requirements.

With kind regards,

Bao Yang, Ph.D, Prof.

Academic Editor

PLOS ONE

Additional Editor Comments (optional):

Reviewers' comments:

Reviewer's Responses to Questions

**Comments to the Author**

1. If the authors have adequately addressed your comments raised in a previous round of review and you feel that this manuscript is now acceptable for publication, you may indicate that here to bypass the “Comments to the Author” section, enter your conflict of interest statement in the “Confidential to Editor” section, and submit your "Accept" recommendation.

Reviewer #2: All comments have been addressed

Reviewer #3: All comments have been addressed

2. Is the manuscript technically sound, and do the data support the conclusions?

Reviewer #2: Yes

Reviewer #3: Yes

3. Has the statistical analysis been performed appropriately and rigorously? 

Reviewer #2: Yes

Reviewer #3: Yes

4. Have the authors made all data underlying the findings in their manuscript fully available?

Reviewer #2: Yes

Reviewer #3: (No Response)

5. Is the manuscript presented in an intelligible fashion and written in standard English?

Reviewer #2: Yes

Reviewer #3: Yes

6. Review Comments to the Author

Reviewer #2: (No Response)

Reviewer #3: (No Response)

7. PLOS authors have the option to publish the peer review history of their article (what does this mean?). If published, this will include your full peer review and any attached files.

Reviewer #2: No

Reviewer #3: Yes: Xiaojian Zhang

---

## [Editor Report · Acceptance letter]

2 Mar 2020

PONE-D-19-22736R2 

Temporal-spatial variability of modern climate in the Altai Mountains during 1970-2015 

Dear Dr. Zhang:

I am pleased to inform you that your manuscript has been deemed suitable for publication in PLOS ONE. Congratulations! Your manuscript is now with our production department. 

With kind regards,

on behalf of

Dr. Bao Yang 

Academic Editor

PLOS ONE